# Mental Illnesses in Inflammatory Bowel Diseases: *mens sana in corpore sano*

**DOI:** 10.3390/medicina59040682

**Published:** 2023-03-30

**Authors:** Bianca Bartocci, Arianna Dal Buono, Roberto Gabbiadini, Anita Busacca, Alessandro Quadarella, Alessandro Repici, Emanuela Mencaglia, Linda Gasparini, Alessandro Armuzzi

**Affiliations:** 1IBD Center, Humanitas Research Hospital-IRCCS, Via Manzoni 56, Rozzano, 20089 Milan, Italy; 2Department of Biomedical Sciences, Humanitas University, Pieve Emanuele, 20072 Milan, Italy; 3Medical Oncology and Haematology Unit, Humanitas Cancer Center, Humanitas Research Hospital IRCCS, Via Manzoni 56, Rozzano, 20089 Milan, Italy; 4Child Neuropsychiatry Unit, Niguarda Hospital, 20162 Milan, Italy

**Keywords:** mental illnesses, inflammatory bowel disease, disability, anxiety, quality of life

## Abstract

*Background and aims*: Inflammatory bowel diseases (IBD) are chronic disorders associated with a reduced quality of life, and patients often also suffer from psychiatric comorbidities. Overall, both mood and cognitive disorders are prevalent in chronic organic diseases, especially in the case of a strong immune component, such as rheumatoid arthritis, multiple sclerosis, and cancer. Divergent data regarding the true incidence and prevalence of mental disorders in patients with IBD are available. We aimed to review the current evidence on the topic and the burden of mental illness in IBD patients, the role of the brain–gut axis in their co-existence, and its implication in an integrated clinical management. *Methods*: PubMed was searched to identify relevant studies investigating the gut–brain interactions and the incidence and prevalence of psychiatric disorders, especially of depression, anxiety, and cognitive dysfunction in the IBD population. *Results*: Among IBD patients, there is a high prevalence of psychiatric comorbidities, especially of anxiety and depression. Approximately 20–30% of IBD patients are affected by mood disorders and/or present with anxiety symptoms. Furthermore, it has been observed that the prevalence of mental illnesses increases in patients with active intestinal disease. Psychiatric comorbidities continue to be under-diagnosed in IBD patients and remain an unresolved issue in the management of these patients. *Conclusions*: Psychiatric illnesses co-occurring in IBD patients deserve acknowledgment from IBD specialists. These comorbidities highly impact the management of IBD patients and should be studied as an adjunctive therapeutic target.

## 1. Introduction

Inflammatory bowel diseases (IBDs), including ulcerative colitis (UC) and Crohn’s disease (CD), are chronic conditions affecting the intestinal wall with a relapsing–remitting behavior [1,2]. The pathogenesis of IBDs remain unknown, but a variety of risks and triggering factors have been recognized, including host genetics, immune dysregulation, and gut microbiota alterations [3]. The natural history of IBD is characterized by periods of quiescence interspersed with episodic flares of disease activity [1,2,3]. IBD markedly impacts the quality of life (QoL) and lifestyle of the affected patients [4], and it is known that patients with IBD might frequently present mental disorders, such as anxiety and depression [5]. However, the biological basis and explanation of the burden of mental illnesses in IBD patients is not fully established, and it is still unclear to what extent these diseases co-occur and in what sequence they mutually develop [6]. The relationship between IBD and psychological factors is a rather old issue: as early as 1930, it was noted that emotional factors and personal experiences were correlated with the severity of the disease, to the point of considering IBDs psychosomatic disorders [7]. The effects of living with the diagnosis of CD or UC on the patients’ psychosocial sphere and QoL has been assessed throughout the last decades [8].

Indeed, in IBD patients, the worsening of the QoL is mainly attributable to the juvenile onset, the chronic nature of the disease, and the unpredictable severity of symptoms and their significant impact on social life [7,8]. These factors have a detrimental effect on the ability of the affected person to carry out routine daily activities, resulting in lower working capacity, a higher likelihood of being unemployed, and diminished economic capacity [9]. Therefore, on the one hand, the diagnosis of IBD itself significantly impacts the mental health of the subject; on the other hand, once intestinal disease develops, the unpredictability, uncertainty, and chronic course of the disease can lead to additional consequences for the patient, such as social isolation, stigmatization, shame of one’s condition, dissatisfaction with one’s body image, and a feeling of poor body hygiene and sexual inadequacy [9,10].

In the management of IBD, psychological distress deserves particular consideration since it does not only affect patient’s QoL, but is also associated with increased disease activity, higher frequency of relapse, and greater use of health services [11].

It has been recognized that the individual response to stress depends on the personal perception of the stressful event, which is more heightened in subjects with IBD [7,12]. In fact, several studies have demonstrated that psychological distress and its personal perception by IBD patients can influence worse health outcomes [12]. In a prospective study of patients with UC in clinical remission, long-term perceived stress tripled the risk of exacerbation of disease [13], and, interestingly, this association remained after adjusting for possible confounding factors (i.e., shorter sleep, shorter remission) [13].

Among psychiatric illnesses, anxiety and depression are the disorders most frequently associated with IBD [3]. In the attempt to causatively link IBD and psychiatric disorders, a great body of research focused on the central role of the gut–brain axis [6].

In the last years, the relationship between mental illness and IBD has received considerable interest; observational studies have shown that pre-existing psychiatric morbidity is associated with adverse outcomes during IBD longitudinal follow-up, and the inflammatory activity is associated with the de novo development of psychological disorders [14]. Psychological stress has been addressed as a possible cause of the altered permeability of the intestinal mucosa, with subsequent cytokine secretion [15]; this mechanism possibly influences the risk of relapse and disease severity of IBD. Among the potential predictors of mental illness, a history of previous surgery, female gender, extra-intestinal manifestations, and the use of tobacco have been identified [16].

Moreover, patients with IBD are probably more vulnerable to the effects of stress; the identification of potential markers of individual vulnerability to stress for the use of psychological interventions to reduce stress appears to be an accessible therapeutic strategy [17].

Whether gastrointestinal inflammation favors the development of mental illnesses, or rather the opposite, remains an open question that will be addressed in our review. In this review, we aim to summarize the current evidence on neural interactions driven by intestinal inflammation and to examine the burden of mental illnesses in IBD.

## 2. Materials and Methods

We searched PubMed until January 2023 using the following terms: “Inflammatory Bowel Diseases”, “Crohn Disease”, “Ulcerative Colitis”, “Depression”, “Anxiety”, “Mood Disorders”, “Stress”, and “Cognitive disorders” to recognize relevant publications exploring the implications of the brain–gut axis in patients with IBD, the association between mental illnesses and IBD, and any related therapeutic applications. Both animal and human studies were included.

## 3. Neural Mechanisms Implied

The mutual association between IBD and psychological illnesses can be explained by a bidirectional communication via the gut–brain axis. The term “gut–brain axis” refers to the complex interactions between neuroendocrine pathways, the central nervous system (CNS), the peripheral nervous system, and the gastrointestinal tract through the enteral nervous system, as well as paracrine regulations [18,19]. Mental disorders, especially depression, partially share some pathophysiological mechanisms of IBD, including oxidative stress, increased proinflammatory cytokines and C-reactive protein (CRP), dysbiosis, and gut permeability [20]. Moreover, many inflammatory signaling pathways (i.e., IL-1–6, IL-23, and CRP) depend on a parasympathetic and hypothalamus–pituitary–adrenal (HPA) axis dysregulation, which has been proven to participate in triggering mental disorders, including depression, schizophrenia, and bipolar disorders [20,21].

In detail, increased levels of stress induce the activation of the HPA axis, favoring the release of the corticotropin-releasing factor (CRF), which in turn stimulates the anterior pituitary gland to release adrenocorticotropic hormone (ACTH) with direct effects on the intestine, such as an increase in intestinal permeability [22].

At the same time, stress promotes the activation of the sympathetic nervous system, resulting in the release of catecholamines from the adrenal medulla and a reduction in parasympathetic activity [23,24,25,26]. These factors lead to a massive secretion of proinflammatory cytokines, determining inflammation in the intestinal tract [22,23,24,25].

In addition, the immune system itself can affect afferents of the vagus nerve, and circulating cytokines can induce the cerebral production of prostaglandins and nitric oxide, activating leukocytes to enter the brain through circumventricular organs [26,27].

The vagus nerve performs a modulating action of inflammation through different pathways, making the vagal nerve stimulation a catching point in the management of IBD [28]. The role of the HPA axis, the importance of catecholamines in inflammatory pathways, and the vagus nerve action underline the importance in the connection between inflammatory illness and neural mechanism [28].

In detail, Ghia et al. demonstrated that, in mice, depression induces an exaggerated response to inflammatory stimuli in the gut and increases susceptibility to intestinal inflammation thorough the impairment of the parasympathetic system (i.e., tonic vagal inhibition) [29]. Vagotomized mice displayed severe colitis, and, after the administration of tricyclic antidepressants, a restoration of the vagal function was observed, as well as a reduced intestinal inflammation [29].

Summarizing these data, we can conclude that both acute and chronic stress, to which IBD patients are exposed, during their disease course increases intestinal permeability, weakens tight junctions, and increases bacterial translocation through the intestinal wall [30]. Several studies investigating the causative mechanisms shared by intestinal inflammation and psychiatric illnesses have been conducted in mouse models.

It was demonstrated that in adult male mice models of colitis induced by intrarectal injection of DNBS (dinitrobenzene sulfonic acid), depressive and anxious behaviors were associated with increased expression of inflammatory genes and abnormal mitochondrial function in the hippocampus [31]. These results suggest that the peripheral inflammation can, to some extent, increase the transcriptional levels of the genes in the toll-like receptor pathway, and these negative effects may be involved in the co-occurrence of anxiety and depression in the early stages of colitis, especially CD [31].

Furthermore, a study by Carloni et al. identified the presence of a vascular barrier of the choroid plexus in the CNS, which closes during acute phases of intestinal inflammation as a defense strategy through the wingless-type, catenin-beta 1 (Wnt/β-catenin) signaling pathway, explaining the behavioral change consistent with depression and anxiety in mice after induced colitis [31]. This mechanism leads to the production of pro-inflammatory cytokines, especially in the hippocampus [32].

Additional evidence on the impact of inflammation on the CNS derives from recent studies on dextran sodium sulfate mouse IBD models: it was demonstrated that chronic intestinal inflammation, through increased plasma levels of IL-6 and TNF-α, modifies and reduces the neurogenesis, specifically in the hippocampus [33]. In similar experiments, it was shown that numerous inflammatory-related genes (TGF-β, Smad-3, IL-6, IL-1β, and S-100) are upregulated in the CNS, causing microgliosis and astrocyte activation [34].

These studies endorse an impaired neurogenesis and an altered CNS homeostasis as a possible biological basis of psychiatric illnesses in patients with IBD.

Several observations have also been made in human studies, mainly derived from neurofunctional studies. In a study including 74 patients affected by UC investigating global and local networks with functional brain imaging, a significantly lower neural modularity was observed compared to healthy controls (*p* = 0.015), as well as significantly enhanced connectivities in somatomotor, dorsal attention and the visual subnetwork in UC patients compared to healthy controls [35]. Major differences in structural brain measures have been described in patients with CD, even those in clinical remission, as compared to age- and gender-matched controls, such as an increased average left hemisphere cortical thickness (mean, 2.68 mm ± 0.17 SD, *p* < 0.01), including in the left superior frontal region, a functional area implicated both in cognitive and affective processes [36]. In Figure 1, the interactions between chronic intestinal inflammation and the CNS are presented.

Figure 1 shows the bidirectional communication between the gastrointestinal tract and the central nervous system via the gut–brain axis, vagus nerve, and hypothalamus–pituitary–adrenal axis.

## 4. Incidence and Prevalence

Overall, both mood and cognitive disorders present higher rates of prevalence among patients affected by IBD compared to a healthy population [37]. Additionally, higher rates of mental illnesses are reported, particularly in concomitance with flares of disease activity [37]. Below, we report summarized epidemiological data of psychiatric comorbidities in IBD divided into subgroups. Table 1 shows the main data on incidence and prevalence of mental illnesses in IBD.

### 4.1. Depression

Estimated prevalence rates, derived from systematic reviews and cohort studies, of depressive disorders among patients with IBD range from 15% to 40% [38,39]. Notably, a significant association between disease activity and both depression and anxiety are reported (*p* = 0.01) [40]. The prevalence of psychiatric illnesses in IBD has been examined in different studies with different methodologies: many studies adopted the International Classification of Diseases 9 (ICD-9) or ICD-10 codes to evaluate depression and/or anxiety, while most studies employed several questionnaires, among which the most common is the Hospital Anxiety and Depression Scale (HADS).

Regarding the prevalence, currently robust data on the prevalence of depression in the IBD population have been endorsed by three meta-analyses, where a pooled prevalence of depressive symptoms ranged from 21.0% to 25.2% [5,37,38], with recurring observation of a higher prevalence in patients with active intestinal disease [5,39,40].

In a recent nationwide, population-based study, Choi et al. estimated an incidence rate of depression of 14.99 per 1000 persons/year in the CD patients’ group, while for UC, an incidence rate of 19.63 per 1000 persons/year was assessed, both significantly higher compared to the non-IBD controls (*p* = 0.01) [41]. The authors reported, over a mean follow-up of 6 years, a cumulative incidence of depression of 8% vs. of 4% in unaffected controls [41]. Further data have shown depression incidence rates of 0.89 (0.84–0.95), and 1.61 (1.48–1.75) over a period of 5 years after the diagnosis of IBD and over a mean follow-up of 9.6 years, respectively [42,43]. Additional data worth mentioning concern the high prevalence of suicidal ideation, suicide attempts, and suicide: the risk appears higher in specific groups (i.e., Crohn’s disease subtypes, female IBD, pediatric-onset IBD, young adult IBD, and elderly-onset IBD). In addition, suicide itself appears to be more strongly associated with CD [44,45,46].

### 4.2. Anxiety

According to current evidence, anxiety is the most frequently associated psychiatric condition in patients with IBD [5,39,40]. An incidence rate of anxiety of 20.88 per 1000 persons/year in CD patients and an incidence rate of 31.19 per 1000 persons/year in UC patients were assessed vs. 14.31 in non-CD controls and 21.55 in non-UC controls, respectively (*p* = 0.01) [41]. As reported in meta-analyses, the pooled prevalence of anxious symptoms varies from 19.1% to 32.1% [5,37,38]. Concerning lifetime prevalence, in the Manitoba IBD cohort study, markedly higher rates of anxiety disorders among IBD patients were found. More precisely, lifetime prevalence of major depression was assessed at 27.2% (vs. 12.3% in healthy controls, OR 2.20, 95% CI 1.64–2.95), and the lifetime prevalence of any anxiety or mood disorder was assessed as 35.8% (vs. 22.1 in healthy controls, OR 1.24, 95% CI 0.96–1.59) [47]. Additionally, the same study reported that, for patients with IBD and anxiety, in around 80% of the cases, the first episode of anxiety anticipated the diagnosis of IBD by 2 years or more [47].

### 4.3. Bipolar Disorder

Bipolar disorder is defined by irregular cycles of mania, hypomania, depression, and/or mixed mood states, proven to be rather prevalent in IBD patients [48]. Data from an Asian population-based cross-sectional study show that bipolar disorder was more frequent in IBD patients than in matched comparison patients without IBD, and that the adjusted odds ratio of IBD patients developing a bipolar disorder was 2.10 (95% CI 1.30–3.38) [49]. Moreover, with specific respect to UC patients, the adjusted odds ratio appeared higher (OR 2.23, 95% CI 1.31–3.82) compared to the comparison population [49]. Similarly, the estimated incidence of bipolar disorder in a Canadian population-based study was significantly higher in the IBD group compared to the matched cohort (IRR, 3.80, 95% CI 2.29–6.30; vs. 1.56, 95% CI 1.09–2.23), with concordant higher incidence rate ratios (IRR, 1.82; 95% CI, 1.44–2.30) [50].

### 4.4. Cognitive Disorders

The term ‘cognitive’ refers to thought and numerous related processes; the corresponding disorders are characterized by an acquired impairment to one or more among learning and memory, complex attention, language, pre-conceptual motor functions, executive functions, and social cognition [51]. Cognitive impairment has been suggested as a potential extraintestinal manifestation of IBD. Indeed, data derived from meta-analyses have shown that IBD patients exhibit objective deficits in attention and executive functions, especially in working memory, compared with healthy controls [52,53]. Nevertheless, the cognitive impairment in IBD appears to be less frequent than mood disorders, and mild compared to other chronic conditions (i.e., multiple sclerosis) [52,53].

As shown in a recent cross-sectional multicenter study including CD patients, the disease activity was linearly correlated with age- and education-adjusted cognitive function scores [54]. As revealed by Ascertain Dementia 8 (AD8) and MFI questionnaires, approximately 50% of the included patients reported that their subjective cognitive capabilities were declining or reported extreme cognitive fatigue [55]; objective cognitive scores below one standard deviation of age and education expected average were then confirmed in nearly 37% of the patients [54]. Crohn’s disease activity index and nutritional risk index significantly correlated with cognitive scores (*r* = −0.34, 0.39, 0.33, *p* < 0.05), and both significances remained independent of associated depression (*p* < 0.05) [4].

Concerning dementia, preclinical models of Alzheimer’s disease (AD) also showed that IBD itself aggravated the AD course [55]. In a population-based cohort analyzing more the 1700 IBD patients and 17,000 controls, an overall higher incidence of dementia, mostly AD, as well as an earlier age of presentation among patients with IBD was observed compared with controls: 5.5% vs. 1.4% and 76 vs. 83 years, respectively [56].

**Table 1 medicina-59-00682-t001:** Data on incidence and prevalence of mental illnesses in IBD. IBD: inflammatory bowel disease; UC: ulcerative colitis; CD: Crohn’s disease.

Reference	Year	Study Design	Observation Time	Incidence/Prevalence of Depression	Incidence/Prevalence of Anxiety	Incidence/Prevalence of Bipolar Disorder	Incidence/Prevalence of Dementia
Barberio et al. [5]	2021	Meta-analysis		Pooled prevalence of depressive symptoms: 25.2%	Pooled prevalence of anxiety symptoms: 32.1%		
Mikocka-Walus et al. [37]	2016	Systematic review		Pooled prevalence of depression in IBD vs. healthy control: 21.2% vs. 13.4%, in active vs. inactive disease: 34.7% vs. 19.9%	Pooled prevalence of anxiety in IBD vs. healthy control: 19.1% vs. 9.6%, in active vs. inactive disease: 66.4% vs. 28.2%		
Neuendorf et al. [38]	2014	Systematic review		Pooled prevalence of depression disorders: 15.2%, depressive symptoms: 21.6%, pooled prevalence of depressive symptoms in active vs. inactive disease: 40.7% vs. 16.5%	Pooled prevalence of anxiety disorders: 20.5%, anxiety symptoms: 35.1%, pooled prevalence of anxiety symptoms in active vs. inactive disease: 75.6% vs. 31.4%		
Choi et al. [41]	2019	Cohort study	6 years	Incidence rate (per 1000 persons/year) in CD vs. non-CD controls: 14.99 vs. 7.75 In UC vs. non-UC controls: 19.63 vs. 11.28 Prevalence in IBD vs. non-IBD controls: 8.0% vs. 3.7%	Incidence rate (per 1000 persons/year) in CD vs. non-CD controls: 20.88 vs. 14.31, in UC vs. non-UC controls: 31.19 vs. 21.55 Prevalence in IBD vs. non-IBD controls: 12.2% vs. 8.7%		
Marrie et al. [42]	2017	Cohort study	9.6 years	Incidence rate ratio of depression in IBD of 1.61	Incidence rate ratio of anxiety in IBD of 1.37		
Walker et al. [43]	2008	Cohort study		Lifetime prevalence of major depressive disorder in IBD vs. control: 27.2% vs. 12.3%	Lifetime prevalence of any anxiety or mood disorder in IBD vs. control: 35.8% vs. 22.1%		
Kao et al. [48]	2019	Population-based cohort study				Adjusted odds ratio of developing a bipolar disorder of 2.10 (95% CI 1.30–3.38)	
Bernstein et al. [50]	2019	Population-based cohort study				Incidence in the IBD group as compared to the matched cohort (3.80, 95% CI 2.29–6.30; vs. 1.56, 95% CI 1.09–2.23), higher incidence rate ratios (IRR, 1.82; 95% CI, 1.44–2.30)	
Zhang et al. [56]	2021	Cohort study	16 years				Incidence (per 1700 persons) of dementia in IBD vs. healthy control: 5.5% vs. 1.4%

Nevertheless, meta-analyses have underlined that, despite the available data on the association of IBD with subsequent dementia development, the exact risk of dementia in IBD cannot be precisely established due to high heterogeneity in the study design of the available studies [57]. Finally, from observational analyses, the causal role of IBD in triggering AD appears unlikely, as it may result from confounding factors, and the evidence remains weak [58].

### 4.5. Schizophrenia

Among patients with schizophrenia, the risk of developing IBD is higher than in the general population (1.14% vs. 0.25%) [59]. The connection between schizophrenia and IBD also has genetic fundamentals, as shown by a Mendelian randomization analysis which provided causal effects of schizophrenia on IBD but not vice versa [60]. Patients with schizophrenia have been demonstrated to show higher gut permeability, defined as a lactulose/mannitol ratio ≥0.1, than controls (22.7% vs. 5.8%, OR 4.8, 95%, CI 1.2–18.3, *p* = 0.03) [61]. In the etiopathogenesis of schizophrenia, the impaired intestinal permeability with subsequent inflammation may have a triggering role [62], and it was shown that the serum zonulin levels, which regulate intestinal and blood–brain barrier tight junctions [63], were significantly higher in patients with schizophrenia than in the unaffected population [64].

## 5. Compliance to Therapy and Role of Psychotherapeutic Intervention

Nonadherence to maintenance therapy, either oral or biologic, in patients with IBD is a significant healthcare problem and can lead to unnecessary therapy escalation.

Overall, medication nonadherence can occur in up to 45% of patients with IBD and is markedly influenced by psychological factors, particularly psychological distress, and patients’ beliefs [65].

Compliance has been repeatedly demonstrated to inversely correlate with the presence of psychiatric disorders in IBD cohorts [66,67]; the co-occurrence of anxiety and mood disorders significantly increased the risk of discontinuation in the first year following anti-TNF initiation (hazard ratio, 1.50; 95% CI, 1.15–1.94) and the overall risk of discontinuation of anti-TNF therapy (adjusted hazard ratio, 1.28; 95%, CI, 1.03–1.59) [67]. Concerning anti-TNF, further data have shown that depressive symptoms at baseline significantly led to noncompliance over a 2-year period of follow-up (HR 2.28, CI 1.1–4.6, *p* < 0.05) [68].

However, the effect of psychiatric disorders on non-compliance is not limited to parenteral administration; indeed, in the early 2000s, it was also observed with amino-salicylates and thiopurines [69,70,71].

Psychological interventions can also address disease acceptance and pain misrepresentation and misreporting. Among the possible psychotherapeutic interventions, cognitive behavioral therapy, hypnotherapy, and mindfulness therapy are included. In a prospective multicenter study, the adoption of structured personalized counseling sessions led to significantly increased medication acceptance rates in nonadherent IBD patients over a follow-up of 24 months [72].

However, the available meta-analyses did not endorse this evidence: a first systematic review with meta-analysis, in 2011, found that psychotherapy had no effect on the QoL, emotional status, or disease activation in adult patients with IBD, while in adolescents, psychological interventions appeared to be beneficial, despite limited evidence [73]. With respect to adolescent IBD patients, in a randomized controlled trial (RCT) investigating the effect of cognitive behavioral therapy on disease course, the time to relapse did not differ between the intervention group and the controls (*p* = 0.915), or in the course of clinical disease activity over time between the two groups [62]. Regardless, the psychotherapy duration significantly affected fecal calprotectin (β, −0.11, 95% CI, −0.195 to −0.031; *p* = 0.008) [74]. Importantly, behavioral interventions have been demonstrated to improve medication adherence in adolescent IBD patients [75].

A later analysis confirmed no effect of individual psychological therapies either on psychological wellbeing scores or on disease activity indices [64], except for cognitive behavioral therapy, which was proven to have small short-term beneficial effects on depression scores and QoL in patients with IBD [76].

Notably, the heterogeneity of psychological interventions and the design of the included studies may have created bias in the results of these meta-analyses, and more RCTs are required to accurately address the role of psychological therapies in the management of IBD.

According to the results of a latter prospective study, among those IBD patients who accepted psychological intervention, the frequency and number of emergency visits significantly decreased over the year after the start of psychotherapy, compared with the year before psychotherapy (*p* < 0.05) [76]. Psychotherapy was associated with a net saving of resources in the cost–benefit analysis [76].

Finally, evidence from studies conducted in operated IBD patients, specifically those undergoing stoma surgery, supports perioperative psychological support in order to reduce patients’ distress and anxiety related to surgery, as well as to ameliorate psychological and surgical outcomes in the postoperative follow-up [77,78].

As for mindfulness strategies, an RCT conducted by Jedel S et al. including patients with UC in remission examined the efficacy of mindfulness-based stress reduction (MBSR) to reduce disease flares and improve QoL [79]. Lower stress and an improved QoL while in active disease were observed compared to flared patients in the control group (*p* = 0.04 and *p* < 0.01, respectively); however, no effect on inflammatory markers and disease course was observed [79]. Further RCTs have reported an improvement in QoL and depression scores in IBD patients under mindfulness-based stress reduction therapy [80,81]; notably, disease course, disease activity, and inflammatory markers of the intestinal disease were not ameliorated [81,82].

## 6. Discussion

What emerges from our review is that the prevalence of mental illnesses in IBD patients is high, ranging from 15% to 40%, particularly anxiety [5,6,37,38,39]. The association between IBD and mood disorders is relevant, while the link with other psychiatric illnesses, such as dementia and schizophrenia, appears lower [50,58].

Considering their high prevalence, mental illnesses can be regarded as proper extra-intestinal comorbidities in these patients (Figure 2) [5,6,37,38,39].

Despite this evidence, psychiatric comorbidities remain under-recognized and inadequately treated. Our review highlights the limitation of systematically summarizing the true magnitude of mental illnesses in IBD due to the lack of prospective, large, and population-based studies.

Concerning the pathophysiology of mental illnesses in IBD, it appears clear that the central role of the vagal nerve is regulating signals from the depressed brain areas to the gastrointestinal tract, and vice versa [26,27,34] (Figure 1). The involvement of the vagal nerve together with the role of inflammatory mediators is also endorsed by pre-clinical data on the development of depression subsequent to induced colitis, and conversely, the occurrence of colitis after induced depressive symptoms in animal models [26,27,34].

Many open questions remain regarding the temporal relationship between IBD and depression and/or anxiety, any possible genetic correlation between IBD and psychiatric diseases, the true benefit of psychological interventions in regard to IBD disease severity, and the possible influence of IBD therapies on the occurrence of depression and/or anxiety. Since long-term medication is crucial for maintaining remission in IBD patients, a close collaboration between gastroenterologists, psychiatrists, and family members is advised.

Notably, most of the studies have demonstrated the usefulness of psychotherapeutic intervention on quality of life, perception of stress, and adherence to therapies [74,75,76], as well as on more concrete indicators such as emergency visits and saving of resources [77], though it seems reasonable to exclude the true impact of the available psychotherapeutic approaches on disease severity, course, natural history, and surgery rates, where the biology alone mostly impacts. However, there may be advantages in targeting specific subgroups of IBD patients, including youths and adolescents, and those with significant psychiatric comorbidity [74,75,76].

In our view, it would be helpful to immediately integrate a psychiatric consultation and potential psychotherapy into the clinical management of these specific patients, and particular long-term psychological support should be considered for patients with pediatric-onset IBD.

In conclusion, it is essential for dedicated physicians managing IBD patients not only to acknowledge the psychiatric illnesses co-occurring in these patients, but also to consider these comorbidities as part of the medical context and as adjunctive outcomes (Figure 2). The separation between physical and mental health appears misleading and outdated when caring for patients with IBD.

## Figures and Tables

**Figure 1 medicina-59-00682-f001:**
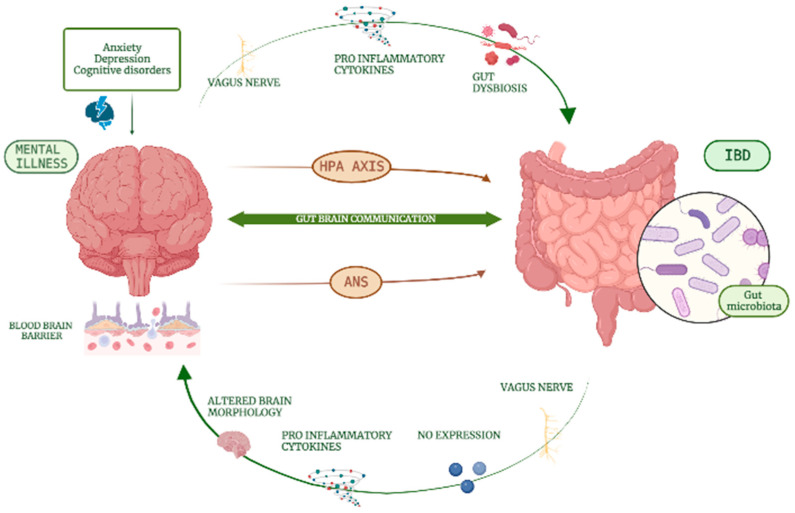
Interactions between chronic intestinal inflammation and the central nervous system. HPA: hypothalamus–pituitary–adrenal, IBD: inflammatory bowel diseases, ANS: autonomous nervous system.

**Figure 2 medicina-59-00682-f002:**
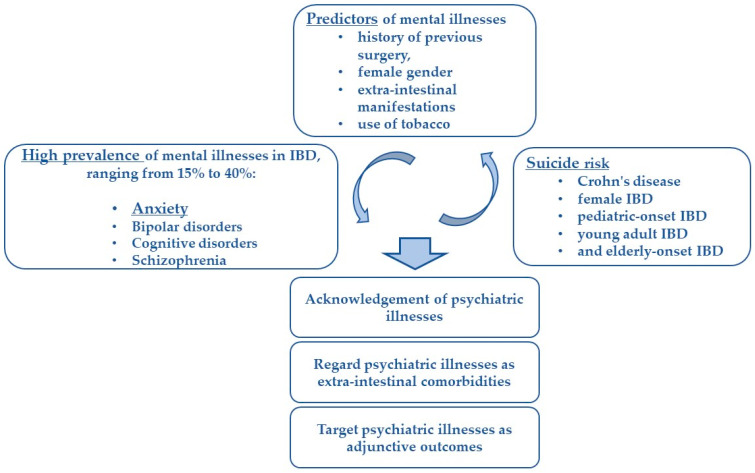
Integration of mental illnesses in the management of IBD.

## Data Availability

No new data were created or analyzed in this study. Data sharing is not applicable to this article.

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
