# Peer review of "Mental Illnesses in Inflammatory Bowel Diseases: mens sana in corpore sano"

_medicina, 2023, doi:10.3390/medicina59040682_

Round 1

Reviewer 1 Report

This review takes an ambitious approach in relating the presence of mental illness in IBD-patients, and points out some possible mechanisms for an interplay. In general the theme is well covered, but needs throughout to be more precisely structured, possible mechanisms highlighted, and, the text here and there shortened.

In the Introduction some important information can be included from two papers:

Maunder RG, Levenstein S. The role of stress in the development and clinical course of inflammatory bowel disease: epidemiological evidence. Curr Mol Med. 2008 Jun;8(4):247-52.

Levenstein S, Prantera C, Varvo V, Scribano ML, Andreoli A, Luzi C, Arcà M, Berto E, Milite G, Marcheggiano A. Stress and exacerbation in ulcerative colitis: a prospective study of patients enrolled in remission. Am J Gastroenterol. 2000 May;95(5):1213-20.

In Neural mechanisms implied (page 2-4), there is a lot of findings listed, but authors can more in detail interpret their referenced data and highlight possible causative mechanisms.

Furthermore, the therapeutic potential of vagus nerve stimulation needs to be mentioned; there are both animal and human data to support such an approach. Please include.

In general, confounding factors are not at all discussed, such as the influence of medications (corticosteroids induce depression; SSRIs can give diarrhea, etc), gender differences and importantly, underlying risk factors for mental illness in IBD-patients (young age? Medically intractable disease? Fistulizing Crohns disease? Repetitive surgery?). Please include.

The topic of schizophrenia in IBD-patients is almost not at all covered, and I propose that a separate subheading deals with it, including e.g the following references:

Gong W, et al. Role of the Gut-Brain Axis in the Shared Genetic Etiology Between Gastrointestinal Tract Diseases and Psychiatric Disorders: A Genome-Wide Pleiotropic Analysis. JAMA Psychiatry. 2023 Feb 8:e224974.

Qian L, et al. Estimation of the bidirectional relationship between schizophrenia and inflammatory bowel disease using the mendelian randomization approach. Schizophrenia (Heidelb). 2022 Mar 28;8(1):31.

Sung KY, et al. Schizophrenia and risk of new-onset inflammatory bowel disease: a nationwide longitudinal study. Aliment Pharmacol Ther. 2022 May;55(9):1192-1201.

Pouget JG; et al.  Cross-disorder analysis of schizophrenia and 19 immune-mediated diseases identifies shared genetic risk. Hum Mol Genet. 2019 Oct 15;28(20):3498-3513.

Gokulakrishnan K, et al. Altered Intestinal Permeability Biomarkers in Schizophrenia: A Possible Link with Subclinical Inflammation. Ann Neurosci. 2022 Apr;29(2-3):151-158.

Ishida I, et al. Gut permeability and its clinical relevance in schizophrenia. Neuropsychopharmacol Rep. 2022 Mar;42(1):70-76.

Usta A, et al. Serum zonulin and claudin-5 levels in patients with schizophrenia. Eur Arch Psychiatry Clin Neurosci. 2021 Jun;271(4):767-773.

Details

Authors searched PubMed for publications, but do not provide the number of relevant papers included and how their selection was carried out. Please correct.

Page 5, line 226: 3.80 is OR or IRR? Clarify here as well as in Table 1. Line 243: spell out “AD8”.

Page 6, 2nd paragraph, lines 263-266: Move to Discussion.

Page 7, line 323; reference no 68 is incorrectly referenced here.

Page 7, paragraph 9, lines 355-358: Move to 4.7 Depression (page 4).

Page 8, 2nd paragraph, lines 363-366: move to Results.

Table 1: Check the order for the prevalence data from the Mikocka-Walus paper; I guess you have changed the order of ‘active vs inactive disease’ in the columns. Right?

Author Response

Response to the editor(s) of ‘Medicina’

Dear Editor(s),

We sincerely thank you for giving us the opportunity to consider for re-submission and potential publication a revised version of our original manuscript entitled “Mental illnesses in inflammatory bowel diseases: mens sana in corpore sano" by Bartocci et al.

We kindly thank the Reviewers for the precious comments. We are pleased to know that you appreciated the topic of our manuscript. The manuscript has been significantly revised and improved according to the received suggestions. Included below you can find a point-by-point response to the remarks.

Sincerely,

Alessandro Armuzzi, Professor

alessandro.armuzzi@hunimed.eu

Reviewer 1

This review takes an ambitious approach in relating the presence of mental illness in IBD-patients, and points out some possible mechanisms for an interplay. In general the theme is well covered, but needs throughout to be more precisely structured, possible mechanisms highlighted, and, the text here and there shortened.

In the Introduction some important information can be included from two papers:

Maunder RG, Levenstein S. The role of stress in the development and clinical course of inflammatory bowel disease: epidemiological evidence. Curr Mol Med. 2008 Jun;8(4):247-52.

Levenstein S, Prantera C, Varvo V, Scribano ML, Andreoli A, Luzi C, Arcà M, Berto E, Milite G, Marcheggiano A. Stress and exacerbation in ulcerative colitis: a prospective study of patients enrolled in remission. Am J Gastroenterol. 2000 May;95(5):1213-20.

Re: Thank You for Your comment. We added the suggested references and commented on that, accordingly.

In Neural mechanisms implied (page 2-4), there is a lot of findings listed, but authors can more in detail interpret their referenced data and highlight possible causative mechanisms.

Furthermore, the therapeutic potential of vagus nerve stimulation needs to be mentioned; there are both animal and human data to support such an approach. Please include.

Re: Thank You for Your comment. We have added details on potential therapeutic role of vagus nerve stimulation, as suggested.

In general, confounding factors are not at all discussed, such as the influence of medications (corticosteroids induce depression; SSRIs can give diarrhea, etc), gender differences and importantly, underlying risk factors for mental illness in IBD-patients (young age? Medically intractable disease? Fistulizing Crohns disease? Repetitive surgery?). Please include.

Re: Thank You for Your comment. We have added details on risk and confoundings, as suggested.

Topic of schizophrenia in IBD-patients is almost not at all covered, and I propose that a separate subheading deals with it, including e.g the following references:

Gong W, et al. Role of the Gut-Brain Axis in the Shared Genetic Etiology Between Gastrointestinal Tract Diseases and Psychiatric Disorders: A Genome-Wide Pleiotropic Analysis. JAMA Psychiatry. 2023 Feb 8:e224974.

Qian L, et al. Estimation of the bidirectional relationship between schizophrenia and inflammatory bowel disease using the mendelian randomization approach. Schizophrenia (Heidelb). 2022 Mar 28;8(1):31.

Sung KY, et al. Schizophrenia and risk of new-onset inflammatory bowel disease: a nationwide longitudinal study. Aliment Pharmacol Ther. 2022 May;55(9):1192-1201.

Pouget JG; et al.  Cross-disorder analysis of schizophrenia and 19 immune-mediated diseases identifies shared genetic risk. Hum Mol Genet. 2019 Oct 15;28(20):3498-3513.

Gokulakrishnan K, et al. Altered Intestinal Permeability Biomarkers in Schizophrenia: A Possible Link with Subclinical Inflammation. Ann Neurosci. 2022 Apr;29(2-3):151-158.

Ishida I, et al. Gut permeability and its clinical relevance in schizophrenia. Neuropsychopharmacol Rep. 2022 Mar;42(1):70-76.

Usta A, et al. Serum zonulin and claudin-5 levels in patients with schizophrenia. Eur Arch Psychiatry Clin Neurosci. 2021 Jun;271(4):767-773.

Re: Thank you for Your comment. We have improved the text according to Your suggestions. We have added the suggested citations accordingly.

Authors searched PubMed for publications, but do not provide the number of relevant papers included and how their selection was carried out. Please correct.

Page 5, line 226: 3.80 is OR or IRR? Clarify here as well as in Table 1. Line 243: spell out “AD8”.

Re: Thank You for Your comment. We corrected accordingly.

Page 6, 2nd paragraph, lines 263-266: Move to Discussion.

Re: Thank You for Your comment. We corrected accordingly.

Page 7, line 323; reference no 68 is incorrectly referenced here.

Re: Thank You for Your comment. We corrected accordingly.

Page 7, paragraph 9, lines 355-358: Move to 4.7 Depression (page 4).

Re: Thank You for Your comment. We corrected the order accordingly.

Page 8, 2nd paragraph, lines 363-366: move to Results.

Re: Thank You for Your comment. We corrected the order accordingly.

Table 1: Check the order for the prevalence data from the Mikocka-Walus paper; I guess you have changed the order of ‘active vs inactive disease’ in the columns. Right?

Re: Thank You for Your comment. We corrected the order accordingly.

Reviewer 2 Report

The present article delves into a less studied aspect of IBD – mental health. As clinical healthcare professionals it is an aspect, we don’t come face to face with. The article is well structured with adequate English grammar in general. The quality of the information presented is good. Overall, this manuscript can raise awareness concerning the mental status of patients with IBD and encourage healthcare professionals to look more deeply into the topic.

Author Response

Response to the editor(s) of ‘Medicina’

Dear Editor(s),

We sincerely thank you for giving us the opportunity to consider for re-submission and potential publication a revised version of our original manuscript entitled “Mental illnesses in inflammatory bowel diseases: mens sana in corpore sano" by Bartocci et al.

We kindly thank the Reviewers for the precious comments. We are pleased to know that you appreciated the topic of our manuscript. The manuscript has been significantly revised and improved according to the received suggestions. Included below you can find a point-by-point response to the remarks.

Sincerely,

Alessandro Armuzzi, Professor

alessandro.armuzzi@hunimed.eu

Reviewer 2

The present article delves into a less studied aspect of IBD – mental health. As clinical healthcare professionals it is an aspect, we don’t come face to face with. The article is well structured with adequate English grammar in general. The quality of the information presented is good. Overall, this manuscript can raise awareness concerning the mental status of patients with IBD and encourage healthcare professionals to look more deeply into the topic.

Re: Thank You for Your comment. We are glad to know You appreciated our paper.

Round 2

Reviewer 1 Report

The revised paper is improved, but still but needs to be more precisely structured with possible mechanisms and interrelations highlighted.

The authors' interpretation of reported findings are mostly too weak, and suggestions for the clinical practice vague and insufficiently supported by the related studies, i.e. the proposed recommendations tend to be speculative in nature.

Overall the text is too wordy and repetitive; the paper will benefit from a thorough English revision and can be shortened with 25-30% without losing content.

Under "3. Neural mechanisms implied", combine and rewrite the two paragraphs relating to immune functions under vagal tone, i.e paragraph 5 on page 3 and paragraph 5 on page 4.

Author Response

Dear Editor(s),

We sincerely thank you for giving us the opportunity to consider for re-submission and potential publication a revised version of our original manuscript entitled “Mental illnesses in inflammatory bowel diseases: mens sana in corpore sano" by Bartocci et al.

We kindly thank the Reviewers for the precious comments. We are pleased to know that you appreciated the topic of our manuscript. The manuscript has been significantly revised and improved according to the received suggestions. Included below you can find a point-by-point response to the remarks.

Sincerely,

Alessandro Armuzzi, Professor

alessandro.armuzzi@hunimed.eu

Reviewer 1

The revised paper is improved, but still but needs to be more precisely structured with possible mechanisms and interrelations highlighted.

The authors' interpretation of reported findings are mostly too weak, and suggestions for the clinical practice vague and insufficiently supported by the related studies, i.e. the proposed recommendations tend to be speculative in nature.

Re: Thank You for Your comment. We have added Figure 2 to strengthen our interpretation and recommendations.

Overall the text is too wordy and repetitive; the paper will benefit from a thorough English revision and can be shortened with 25-30% without losing content.

Re: Thank You for Your comment. We have shortened several paragraphs as suggested. A native English specking colleague has revised the English language of the paper.

Under "3. Neural mechanisms implied", combine and rewrite the two paragraphs relating to immune functions under vagal tone, i.e paragraph 5 on page 3 and paragraph 5 on page 4.

Re: Thank You for Your comment. We have combined the two paragraphs as suggested.